# If You Aim Higher Than You Expect, You Could Reach Higher Than You Dream: Leadership and Employee Performance

Naveed Ahmad Khan [1,*], Silke Michalk [1], Kirill Sarachuk [1] and Hafiz Ali Javed [2]

1 Faculty of Business Administration, Brandenburg University of Technology, 03046 Cottbus, Germany; silke.michalk@b-tu.de (S.M.); kirill.sarachuk@b-tu.de (K.S.)
2 International Business School, Teesside University, Middlesbrough TS1 3BX, North Yorkshire, UK; chalijaved@hotmail.com
* Correspondence: khannave@b-tu.de

**Abstract:** Leadership is about lifting a person's vision, raising their productivity to higher standards and creating a personality beyond their usual capabilities. Our study examines the Pygmalion effect and leader-member exchange (LMX) on employee task performance via organizational commitment. The data was collected from 280 middle-level managers from top ten banks in Pakistan. This study offers three main contributions to the literature. First, our results discover a strong link between the Pygmalion effect, LMX and employees' task performance through mediation of organizational commitment. Secondly, our study indicates that leaders should have higher quality relationships with employees and higher task performance expectations. Finally, most previous studies focused on leadership approaches or have been conducted in western developed countries. To our knowledge, this study is a unique contribution to the literature on leaders' expectations in changing and diverse environments, as in underdeveloped countries like Pakistan.

**Keywords:** employee performance; leader-member exchange (LMX); leaders expectations; social exchange; commitment

## 1. Introduction

In order to satisfy human needs, organizations develop innovative products and services which may grant them sustainable competitive advantage. To achieve long-term goals, firms use a wide range of resources, including human skills and potential to satisfy the individuals' demands by adhering to basic social standards Shahzad et al. (2018).

Employee performance creates the antagonistic advantage of the organization in current fierce competition and strengthens its long-standing prosperity. If a firm encourages own employees entirely by monetary contract, it will not be able to maximize the workforce productivity. Therefore, a promising social and psychological support is needed Bartlett et al. (2012); Cascio (2003).

There is evidence in existing scientific literature that leaders' high expectations shape individuals' task performance. When perceived by employees, leader expectations (LE) drive employee performance. The definition of leader expectation draws on the individual beliefs of employees about the expectations of leaders regarding their assigned duties with respect to engagement in a particular job. The expectation of occurrence is high when workers behave to favor a particular event to happen—in other words, people perform willingly when they expect something to happen Tierney and Farmer (2004). When leaders show these high expectations, employees are likely to try to meet their expectations in the context of intentions to perform tasks, maintain quality, or work harder.

The reciprocity of social exchange underpins the association between the leader and their follower. The role of the leader is able to enhance reciprocity and to improve relationships, and this effect tends to get stronger over time Emerson (1976). However, expectations

between leader and follower in the context of behavior may differ, resulting in misunderstandings and generating uncertainty. Hence, an enlightenment of the workplace is critical in meeting the expectations of both leaders and followers. Despite the fact that employees are well informed about what they are expected to do—either through written instruction or verbal—most organizations are unaware about the real workload and the enthusiasm of employees Van Hemmen et al. (2015). As observed, when leader performs his business position actively and reaps valued customers, the leaders' high expectations boost staff service performance. On the other hand, leaders' high expectations of their followers play an essential role in shaping their identity Likert (1967).

Furthermore, a leader that establishes high expectations sets high targets and encourages their people to generate higher levels of productivity. The acknowledgment and encouragement from that leader will push them to provide high-quality service Livingston (1969). Still, employees responsible for a high quality of work are more conscious and are often seeking a solution to a current problem. Thus, considerable support is required from leadership due to the sensitivity and vulnerability of this work. In order to communicate their own ideas and assure high performance, individuals require high quality associations or relationships Cole et al. (2002). High quality relationships between followers and leaders must be established as a way to boost work performance: within such interactions leaders provide opportunities and essential resources to employees, thus creating a favorable working milieu Farh et al. (2017).

The literature shows a variety of antecedents of performance such as organizational commitment Rigtering and Weitzel (2013), leader-member exchange (LMX) Woo (2018); Farrukh et al. (2017), and culture Rahmah and Fatmah (2018), to name a few. However, research regarding the process through which different contextual factors affect employee performance is underdeveloped Sinha and Srivastava (2013). Additionally, high quality relationships between leaders and employees play a vital role in shaping high working standards and increasing labor task performance. The successful delegation of individual tasks, information and resource support to employees is able to enhance their motivation, and develop and promote their performance Scott and Bruce (1994).

Therefore, in this study, we integrate leader expectations and leader-member exchange as antecedents of employee performance. Generally, high-quality LMX encourages employees to participate in decision-making and commit to a greater extent to their job; they experience fewer task-related challenges, demonstrate loyalty, responsible attitude, and readiness for more responsibility De Jong et al. (2011). Furthermore, high-quality LMX even aspire to go beyond their agreed-upon job duties, to be more flexible and voluntary in their activities, to build a good work atmosphere and to perform better. On the other hand, employee productivity may be influenced by leaders directly, so more attention should be devoted to the leader-employee interaction in order to develop a better knowledge of service quality Grošelj et al. (2020); Lenka and Gupta (2019).

One further aspect, the organizational commitment, suggests employees are more excited when their sense of commitment to the organization is triggered, so organizational commitment is seen as an antecedent to proactivity. Individuals' creative contribution and passionate actions in performing tasks in teams are the examples of proactivity as a result of their commitment to an organization. Strauss et al. (2009) observed that effective commitment can benefit organizations by directing employees' attachment to a company; there is a strong link between organizational commitment and organizational member proactivity. Employees' commitment to a particular organization can lead to proactive behavior and contribute to that organization's success, so energetic behavior within organizations is becoming increasingly important. However, few studies have investigated this problem so far.

Previous research examines exchange relationships in isolation (Buch, 2015), thus largely disregarding the fact that employees are involved in multiple relationships at work, including employee-organization, employee-team and employee-supervisor relationships Bordia et al. (2010). To advance social exchange theory, it has been advocated recently

that the linkages between different relationships should be incorporated into analysis Bordia et al. (2017). Hence, we acknowledge that the LMX exists with other formal and informal organizational relationships simultaneously and that LMX should not be studied in isolation.

Furthermore, Graen (1976) mentions that the relationships that employees have with their coworkers and supervisors represent two key social relationships at work. However, several unproven ideas still exist in the literature with respect to the problem of whether exchange relationships with supervisors are interconnected Cole et al. (2002), thus suggesting the need to investigate whether the interaction between exchange relationships with leaders is able to boost employees' task productivity.

This study finds that LE (Pygmalion effect) and LMX are positively linked to employees' high performance; moreover, the organizational commitment for performance mediates these linkages. Our contribution to the literature is threefold. First, this study attempts to investigate the effect of LE (Pygmalion effect) and LMX on employees' high performance. The literature in this field is scarce and the theoretical development is weak because traditional collaborative leadership approaches are more relevant to the performance Scott and Bruce (1994). Second, this paper contributes to the literature on the quality of work and may also help practitioners in formulating interventions to foster innovations in organizations that will ultimately lead to better task performance. Finally, most of the past studies have been conducted in western developed countries, while our research contributes to the literature and practice by gathering survey data from a population of middle-level managers in the banking sector of Pakistan, a developing and very diverse country with its own social, economic and environmental peculiarities. Although this study does not include any cultural aspects, we imply that there might be some differences in responses to survey questions because of the cultural background. To our knowledge, this is the first paper studying a link between LE and LMX with the mediation of organizational commitment in a non-western country.

Section 2 briefly explains the concepts examined in this paper and describes literature background and hypotheses. Section 3 presents the data and methodology. Findings are presented in Section 4 and then discussed in the following Section 5. Finally, Section 6 concludes our research with limitations.

## 2. Literature Background and Hypotheses

### 2.1. Leader Expectations (Pygmalion Effect)

When we behave in a way that favors the occurrence of a specific event, we perform willingly when that event is expected to occur. On the other hand, leaders' high expectations from their followers play an important role in defining their role identity.

Historically, there was a belief that individual's expectations might influence the behavior of others. This psychological phenomenon, or so called Pygmalion effect, was first coined by Merton in 1948 to describe someone's proclivity to meet other people's expectations Tierney and Farmer (2004). To put it another way, the Pygmalion effect describes a situation in which other people's expectations of a specific person influence other individuals' performance. This effect has been primarily observed in the educational context (classrooms), with the findings indicating that teachers' expectations influence students' academic performance Chang (2011); Friedrich et al. (2015).

At all levels leaders have a dominating influence on employees Dhamija et al. (2019). The adoption of proactive leadership skills can assist in molding staff behaviors in order to attain the required high level of performance. Employee performance is impacted by a leader's activities because a leader communicates their expectations to their followers, and employees fight to reach these standards Suliman (2002).

If we consider the Pygmalion effect relevant for employee performance, a question formulated by Eden becomes relevant Eden (1984) *if raising teacher expectations improves pupil performance, wouldn't raising manager expectations improve subordinate productivity?* Existing literature confirms the significance of the expectations of leaders: the leader who

expects more assumed a higher-level of performance. However, this statement goes back to general knowledge and very few scholars investigated this theory to study how leaders' expectations affect process and can be used to enhance the output of an organization. In contrast to these assumptions, various Pygmalion mechanism theorists have focused on measuring the productivity level of a firm and the increased effort of individuals in accomplishing work tasks.

Still, appraisal of internal working procedures and the impact of the Pygmalion process are significant. Leaders should be aware of the consequences of their expectations for their staff since these high expectations may have a significant impact on employee performance Goddard (1985). According to existing concepts, a so called Galatea effect is one of the fundamentals of the Pygmalion process. Hence, to reach higher levels of productivity supervisors and managers design improved leadership styles and behaviors for subordinates. As a result of the high expectations and unique leadership attention they are receiving, their employee performs better: whether the majority of performance improvements are driven by the leader's high expectations, it is regarded a significant variable within the Pygmalion process. Thus, the Pygmalion effect, with its emphasis on high expectations and impact on behavior, may provide a strong and valuable framework for investigating important concepts such as employee performance. We believe that the Pygmalion effect phenomenon can play an important role in increasing productivity.

**Hypothesis 1a.** *Leaders' expectations (Pygmalion effect) have a positive impact on improving employees' task performance.*

The organizational commitment is described as an individual's personal feelings about the organization. Commitment is observed when an employee's personal values, priorities and goals harmonize with the organization's objectives.

In a study by Joo and Lim (2009), higher commitment levels were observed among employees and firms where employees' and organizational values are aligned; commitment itself refers to employees' psychological attachment to a firm. Smith and Meyer (2009) mention that *in order to create and sustain a desirable organizational outcome, everyone must be a part of the organization hence obey and respect the organization's norms*, so organizational commitment has a critical and positive impact on employee performance.

Despite the *promising management tool* role of the Pygmalion effect Eden (1984), few studies have been conducted at the organizational level. The latest research addressed the so called self-fulfilling prophecy (SFP) phenomenon, leaving however the problem of inconsistencies and challenges in organizational performance (e.g., individual) in the context of the Pygmalion concept unattended Tierney and Farmer (2004). Still, the importance of Pygmalion mechanism lies within the concept of subordinates' high performance as a result of high expectations and a high level of employees' inspiration to perform well and dedicate themselves to achieve better results.

**Hypothesis 1b.** *Leaders' expectations (Pygmalion effect) have a positive impact on the employee performance linked to organizational commitment that individuals perceive as supportive of high performance.*

*2.2. Leader-Member Exchange*

The relationship between leader and employees is a necessary determinant of work attitude and behavior because leaders are those *who inspire their followers through quality relationships and support by providing them with a friendly culture, which then results in employees producing high-quality work* Surucu and Sesen (2019). An authentic interaction between leaders and employees (or high-quality LMX) increases employees' sense of cohesion and trust, while trust between leader and employees is an important prerequisite for better performance. LMX reduces people's fear and gives them confidence so they can perform above-standard and high-quality work. High LMX levels within an organization make

employees feel secure in the knowledge that if they perform well, their output will be acknowledged, but even they fail, their leader will encourage them. Employees in high LMX are closely associated with their leaders who supply them with more technical support, expertise and knowledge. This knowledge and skill may inspire employees' cognitive processes and encourage them to take an active role in task performance Mumford et al. (2002). Furthermore, employees involved in high-quality LMX tend to encourage their colleagues to behave proactively in order to fulfill job responsibilities: when the observer's perception is stretched towards cognitive equilibrium, those individuals having close relationships with a notable person in the team (e.g., the leader) are more likely to be warmly welcomed by other team members. As a result, the focus member's reputation and trustworthiness within the team will increase Lau and Liden (2008).

LMX plays a vital role in increasing staff productivity by providing effective assistance but also strong support. The growth of leadership caliber may offer a strong foundation for the development of employee attributes within a company, either via direct or indirect links between employee performance and leadership. Existing literature Duan et al. (2017) confirms that LMX has a positive influence on high-quality work, resulting in better performance. The explanation for this interaction is based on Emerson's social exchange theory Emerson (1976) which states that employees feel grateful to match their leaders' efforts by engaging in additional work role behavior. As a result, employees respond by participating in proactive actions that promote the business goals. In addition, LMX may encourage proactive behavior among workers and provide employees with a sense of security Huynh et al. (2019) which may be an important incentive to achieve higher performance levels.

**Hypothesis 2a.** *Leader-member exchange has a positive impact on employee task performance.*

There is a concern that LMX may have an impact on employee performance through organizational commitment. We believe that generating such a commitment through friendly, supportive and high-quality relationships with employees cultivates the intention to do work. Kozlowski and Doherty (1989) mention that supervisors and leaders are representatives of organizational policies, strategies and procedures; thus, employees consider their supervisors' actions to be corporate policies. As a result, any encouraging action taken by a leader is regarded as an organization's support which creates a sense of affiliation to the firm. However, even though a relationship between LMX and organizational commitment has been studied so far, how organizational commitment mediates the relationship between LMX and employees' performance is still poorly investigated.

**Hypothesis 2b.** *Leader-member exchange is positively linked to employee performance, provided that organizational commitment is supportive of higher employee performance.*

*2.3. Organizational Commitment and Employee Performance*

One more literature record refers to the positive impact of organizational commitment on employees' performance Khan et al. (2010). An organizational commitment is defined as a cognitive interpretation of organizational values Smith and Meyer (2009) and refers to a psychological relationship.

Provision of essential resources to realize an idea is critical for commitment, and their distribution is another indicator to the organization of the assistance of high-quality employee relationships. The relationship between high quality LMX, commitment and employee performance is also rooted in social exchange theory Emerson (1976): when leaders interact in a trustworthy and appreciative manner, generalized reciprocity appears. The social exchange theory's principle of restricted reciprocity can be used to explain how organizational commitment leads to high employee performance. Employees' commitment to the organization creates reciprocity in the context of intentions to perform tasks, quality work, or more work. As a result, we believe that in reciprocity employees become more

productive and have better intentions to work; this is the stage at which employees feel committed to the organization and perform well.

**Hypothesis 3.** *Organizational commitment have a positive impact on employee performance.*

All five hypotheses are represented by a single model depicted in Figure 1. It also contains a framework along with hypothesized relationships between observed variables.

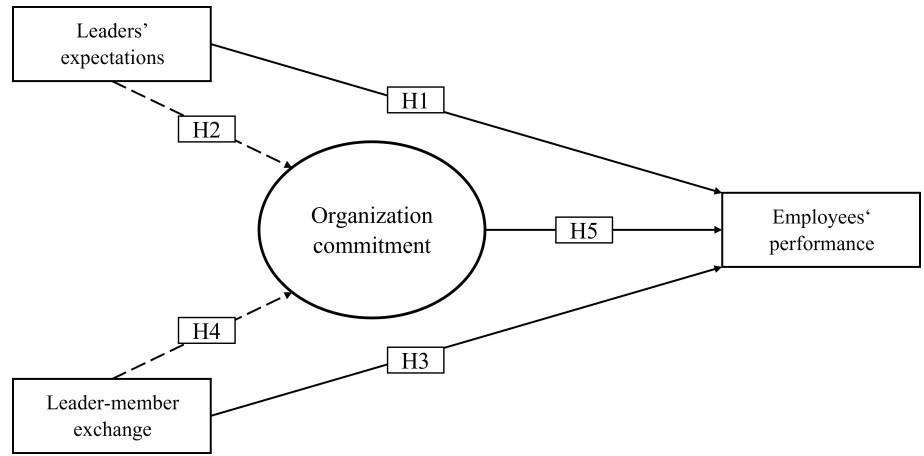

**Figure 1.** Summary of research framework.

## 3. Data and Methodology

While existing research on employees' quality of work and its antecedents have been undertaken from a western viewpoint, in our paper we would like to investigate the influence of leaders' expectations and leader-member exchange in motivating employee performance in an environment which is different from Europe and the US in terms of economy and environmental and social aspects. Pakistan was chosen for our research due to its diverse and collectivist nature Khan (2022) so we expect to record those cultural attributes that may impact the context of LMX and leadership expectations. As long as cultural values act as normative rule guides for employees, dictating the guiding rules and norms in both personal and professional life Walumbwa et al. (2007), individual values show how anyone is impressed and influences others, as well as how leadership is considered and evaluated. Consequently, by targeting the case of Pakistan, this study adds to the literature by considering the role of leadership in employee performance.

The target population for this research includes all employees at the middle level of management (branch, area and operational managers) in the top ten banks, including public, private and multinational corporations, in Pakistan over a ten year period (2010–2020). We focused on middle-level managers because they have a double role—on the one hand, they are leaders to their juniors and, on the other hand, they act as employees to the senior level management. With 8761 branches in the country (for all top ten banks), nearly a third of them are located in three major cities of Pakistan—Karachi, Lahore and Islamabad—with almost ten thousand employees.

For the purpose of this study, the data was gathered at the primary level by a questionnaire. Our questionnaire includes two main sections. First, employees were asked to answer four demographic questions (gender, age, experience, qualification). In the second stage, employees expressed their opinion on over 21 questions covering four major topics:

- employee performance (EP), including five measures of the productivity level (related to high quality, excess, intention, intentions to perform task of work and high level of performance) similar to Shahzad et al. (2018),
- organizational commitment (OC), covering six items as in study by Meyer et al. (1993),
- leader's expectations (LE), with three items adapted from Tierney and Farmer (2004) and

- leader-member exchange (LMX), covering seven items from Scandura and Schriesheim (1994).

The responses from the target audience were gathered using a seven-point Likert scale ranging from strongly disagree (1) to strongly agree (7). In order to fit the questionnaire to the Pakistan context, the wording for some of the questions was changed. After that, the questionnaire was distributed to 650 mid-level management employees (branch managers, area managers and operations managers). A total of 350 filled questionnaires were returned (response rate 54%). After screening, 70 responses were dropped, leaving us with 280 valid responses to analyze (valid response rate 43%).

## 4. Results

### 4.1. Pilot Test

As our questionnaire items were adopted from previously published studies, we carried out a pilot test to guarantee the validity and reliability before a broad distribution of the questionnaires. The pilot test was carried out on a sample of 39 responses, with a minimum suitable threshold of thirty entries for the pilot run as stated by Johanson and Brooks (2010). To test the validity of items, the factor analysis (multivariate technique) was used. Data are subjected to two requirements prior to factor analysis: Kaiser Meyer-Olkin (KMO) test and Bartlett's test of sphericity. The KMO value for our sample is 0.899, which is substantially higher than a threshold value of 0.60 Kaiser (1974).

Bartlett's sphericity test determines if the correlation matrix is an identity matrix, indicating that the factor model is inapplicable Malhotra et al. (2006). If the Bartlett value is significant ($p = 0.05$), then it is possible to employ the principle component analysis (PCA), a process used to compress a larger collection of variables into smaller ones to optimize the interpretation and minimize the information loss Jolliffe and Cadima (2016). For our case, the null hypothesis was rejected (approx. chi-square: 1 221.46; DF = 210; $p = 0.000$) indicating that the variables in the population correlation matrix were uncorrelated. Table 1 contains the factor loadings for the pilot test (items with loadings at 0.40 or less were deleted from the final survey questionnaire).

**Table 1.** Measurement model evaluation for validity and reliability.

| Construct | Items | Factor Loadings | CR | AVE |
|---|---|---|---|---|
| **Employee performance** | EP1 | 0.898 | 0.949 | 0.788 |
| | EP2 | 0.874 | | |
| | EP3 | 0.884 | | |
| | EP4 | 0.896 | | |
| | EP5 | 0.886 | | |
| **Organizational commitment** | OC1 | 0.884 | 0.967 | 0.831 |
| | OC2 | 0.927 | | |
| | OC3 | 0.908 | | |
| | OC4 | 0.910 | | |
| | OC5 | 0.930 | | |
| | OC6 | 0.910 | | |
| **Leaders' expectations** | LE1 | 0.899 | 0.937 | 0.832 |
| | LE2 | 0.912 | | |
| | LE3 | 0.925 | | |
| **Leader-member exchange** | LMX1 | 0.826 | 0.942 | 0.700 |
| | LMX2 | 0.851 | | |
| | LMX3 | 0.890 | | |
| | LMX4 | 0.885 | | |
| | LMX5 | 0.854 | | |
| | LMX6 | 0.764 | | |
| | LMX7 | 0.871 | | |

*4.2. Data Analysis*

Given that the purpose of this study is to predict employee performance based on the leaders' expectations and leader-member exchange, the partial least square structural equation modeling (PLS-SEM) technique was used. PLS-SEM is a common approach for management and social science research which has been widely applied in prior studies Aydin (2020); Khan et al. (2020); Khan (2022) when the major purpose of the study is to assess a core model and show a target construct Hair et al. (2019). Additionally, it prioritizes the optimization of the endogenous construct prophesy above the model fit.

The results were computed with the help of SmartPLS Software (3rd version). PLS-SEM is a two-step process. In the first stage, the measurement model's validity and reliability are evaluated. Table 1 displays the composite reliability (CR), extracted average variance (AVE) and factor loadings (FL). The observed values were above the 0.70 cut-off threshold for factor loadings and composite reliability, and above the 0.50 cut-off point for average variance Hair et al. (2017), thus confirming the measurement model's internal consistency and convergent validity.

We analyze the measurement model's discriminant validity (DV) using the Fornell-Larcker criteria and the heterotrait-monotrait (HTMT) correlation ratio, as indicated by Hair et al. (2017). The degree to which one construct differs from another is referred to as discriminant validity. The observed HTMT ratio for all variables was below 0.85; our model also meets the Fornell-Larcker criterion, demonstrating discriminant validity (see Table 2).

The structural model is examined in the second stage of PLS-SEM evaluation. The bootstrapping tool integrated in SmartPLS Software was used to assess the relevance of all path coefficients. The results displayed in the Table 3 demonstrate that all five hypothesized relationship were supported. The graphical representation of our outcome is depicted in Figure 2.

**Table 2.** Heterotrait-monotrait (HTMT) ratio and Fornell-Larcker criterion.

| HTMT Ratio | EP | LMX | LE | OC |
|---|---|---|---|---|
| Employee performance | | | | |
| Leader-member exchange | 0.794 | | | |
| Leaders' expectations | 0.722 | 0.654 | | |
| Organizational commitment | 0.839 | 0.771 | 0.803 | |
| **Fornell-Larcker Criterion** | **EP** | **LMX** | **LE** | **OC** |
| Employee performance | 0.888 | | | |
| Leader-member exchange | 0.741 | 0.837 | | |
| Leaders' expectations | 0.662 | 0.597 | 0.912 | |
| Organizational commitment | 0.795 | 0.731 | 0.747 | 0.912 |

**Table 3.** Findings.

| Hypothesis | Mean | SD | T-Value | *p*-Value | 95% CI | |
|---|---|---|---|---|---|---|
| **LE » EP** | 0.115 | 0.052 | 2.225 | 0.027 | 0.338 | 0.546 |
| **LE » OC** | 0.480 | 0.058 | 8.292 | 0.000 | 0.013 | 0.217 |
| **LMX » EP** | 0.336 | 0.069 | 4.822 | 0.000 | 0.206 | 0.470 |
| **LMX » OC** | 0.447 | 0.055 | 8.051 | 0.000 | 0.338 | 0.546 |
| **OC » EP** | 0.464 | 0.077 | 6.070 | 0.000 | 0.309 | 0.602 |
| **Organizational commitment mediation** | | | | | | |
| **LE » OC » EP** | 0.223 | 0.050 | 4.490 | 0.000 | 0.135 | 0.325 |
| **LMX » OC » EP** | 0.226 | 0.040 | 5.185 | 0.000 | 0.132 | 0.287 |

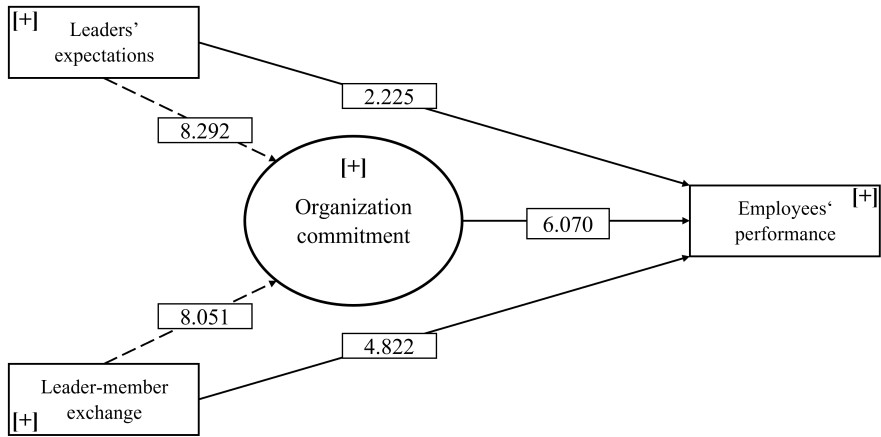

**Figure 2.** Summary of research framework.

## 5. Discussion

This study developed and tested a model to prove that leadership influences employee performance through organizational commitment. We were able to confirm all five hypotheses formulated in this research (see Table 3).

The findings of Hypothesis 1a confirm a significant impact of leaders' expectations of high task performance from their employees and are in line with previous papers linking leaders' high expectations with employees' extra-role behaviors such as high quality of work Farrukh et al. (2019). This demonstrates the importance of individual commitment to the organization in fostering employee high performance. Thus, we may testify that when a leader expects high levels of task achievement from his or her employees, the employees will reciprocate by being involved in the tasks that have been assigned to them.

We also confirmed that employees who have a positive relationship with their boss perceive organizational dedication to be beneficial for better levels of performance (Hypothesis 1b). Furthermore, if the leader has high expectations of their followers in terms of extra-role conduct while simultaneously demonstrating respect to their subordinates, employees will reciprocate by performing high-quality work. This outcome contradicts with the finding by Sutton and Woodman (1989) but, on the contrary, stays in line with the findings in the literature Redmond et al. (1993); Tierney and Farmer (2004); Tierney et al. (1999). These studies demonstrate that the Pygmalion effect may be used to understand employee performance, particularly when the performance criteria is based on high quality of job performance.

Moreover, the results also revealed a positive impact of organizational commitment impact on employee task performance (Hypotheses 2b and 3). Those employees who are committed with the organization yield higher productivity as compared to those who are less committed. We also observed that commitment to the organization mediates not only the relationship between leader-member exchange, but also the relationship between a leader's high expectation (Pygmalion effect) and individual's performance. The leader's high quality relations exchange, as a form of social exchange, generates positive emotions towards the organization in employees, but also generates obligations to the organization.

The results also showed that leader-member exchange (Hypothesis 2a) has a positive impact on employee performance, implying that the performance may be improved by creating a high-quality linkage. We already mentioned that a high-quality connection provides employees with a sense of belonging, independence and better self-esteem, which is reciprocated by employees in the form of extra-role behavior. A clear comprehension of each other's duties may be recognized as a result of a quality exchange link between employees and supervisors, resulting in a greater intention to perform work Atitumpong and Badir (2018).

## 6. Conclusions

Our study adds to the literature and practice in many ways. To our knowledge, this is the first study to look into how leader expectations (Pygmalion effect), leader-member exchange and organizational commitment affect employee performance. Also, although some proof of an association between leader-member exchange and employee performance existed in the literature Atitumpong and Badir (2018), no study examined this association with employee performance, a wide concept constituent of work intentions to perform tasks, excess work and high quality work. Hence, this is a one-of-a-kind contribution to the literature on leadership and performance enhancement. Furthermore, the majority of previous studies on performance have focused on leadership approaches rather than specifically on leaders' expectations (Pygmalion concept). This study broadens our understanding of leaders' expectations in fostering employee extra-role behavior Kierein and Gold (2000); Tierney and Farmer (2004): if employees view organizational commitment to be helpful, leaders' expectations will yield greater results.

Our findings also provide credence to the behavioral plasticity theory Brockner (1988). Employees may feel more valued in the company if they watch a leader's encouraging conduct; this may increase their self-esteem in the organization, prompting them to exert more effort to improve their productivity, especially when followers lack confidence in themselves. Employees with poor organizational self-esteem feel that their efforts would go unnoticed. According to our research, employees with low self-esteem are more impacted by their social circumstances than those with strong self-esteem. This point of view was supported by a number of other scholars as well Mossholder et al. (1981); Gardner and Pierce (1998).

Some of our findings rooting in leader-member exchange theory may be important for managerial purposes. First, we observed that a positive relationship between leaders and members would motivate employees and their interest in non-regular activities, because excess work, intentions to perform tasks and high-quality work are not always part of the official daily routine. As a result, we may recommend managers to cultivate a relationship based on mutual trust, encouragement, empowerment and respect, which may be rewarded with higher performance quality. Additionally, leaders or supervisors have to maintain a high level of interaction with their subordinates by holding frequent meetings for feedback and information exchange which stimulates and encourages subordinates to do high-quality work. Furthermore, firms should encourage and educate managers on how to sustain a high-quality exchange connection by providing them with the required resources.

Our findings support the assumption that employees require a supportive atmosphere in order to execute extra-role activities, and we advise that leaders should pioneer the formation of a pleasant, supporting environment by providing necessary resources and time. Furthermore, our outcome shows that the presence of devoted subordinates increased the influence of leaders' expectations. As a result, we believe that leaders should set extra-role behavior standards for their followers while simultaneously generating an enthusiastic environment in which people desire to remain self-attached.

Although this work credits several significant implications for current theory and practice, a few limitations apply that hint to future research opportunities. We advise that consecutive studies should focus on longitudinal data collection methodologies to better understand the cased and effect connection, because the variables in this study are behavioral and perception-based which may not fairly assess more than one moment in time. Furthermore, because this study did not examine the influence of leader-member exchange and leaders' expectations on dimension level, we urge that future research may investigate this relationship at both the dimension and construct levels for a more comprehensive understanding. Finally, in our paper we tested the outcome for only a few cities, so we may propose testing the same model on a cross-sectional level to improve its generalizability.

**Author Contributions:** Conceptualization, N.A.K. and S.M.; methodology, N.A.K.; software, K.S.; validation, N.A.K. and H.A.J.; formal analysis, N.A.K.; investigation, K.S.; resources, H.A.J.; writing original draft preparation, N.A.K. and S.M.; writing review and editing, K.S.; visualization, H.A.J.; supervision, S.M.; project administration, N.A.K.; funding acquisition, S.M. and K.S. All authors have read and agreed to the published version of the manuscript.

**Funding:** This research received no external funding.

**Conflicts of Interest:** The authors declare no conflict of interest.

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
