# Peer review of "If You Aim Higher Than You Expect, You Could Reach Higher Than You Dream: Leadership and Employee Performance"

_economies, doi:10.3390/economies10060123_

Round 1

Reviewer 1 Report

The major issue for this particular manuscript is that it lacks a solid theoretical foundation.  There are simply concepts that are put together as a framework.  This is the reason the discussion of the findings section is shallow.  To improve this manuscript, my suggestion is to find some leadership theories to support your framework.  Also, each construct needs to be justified for inclusion.  For example, why organizational commitment?  These days, organizational citizenship behavior is more adapted/adopted in the leadership literature.  So, why don't you include organizational citizenship behavior, and not organizational commitment?  This needs to be justified.

Once you include more supporting theories and theoretical concepts in the literature review section, you will be able to draw upon those theories and theoretical concepts to compare and highlight the contribution of your findings. 

Another section needed is managerial/policy implications of your findings.  How do they make managers/leaders more effective as a manager or leader?

In my view, the constructs you used as part of your framework are not new.  They have been around for quite some time.  Therefore, you need to justify the originality of your study.

Author Response

In theoretical background, we have highlighted the research gap and focused on Pygmalion effect variable; the rest of the variables are explained as supporting variables. In context of theories, we took the Pygmalion phenomena, social exchange theory and LMX as itself are separate theory. By considering these theories, we can justify the relationship of expectations and commitment.

Then, the constructs were put in a systematic and logical sequence. Additionally, we decided not to introduce our constructs one by one but combined them together with our hypotheses’ arguments as suggested. The hypotheses were reorganized to a suggested pattern (for example, H1a, H1b etc.). In context of your valuable suggestion to moderate hypotheses, unfortunately we cannot add at this stage because that will modify our results. However, we are considering this moderation in our coming study. The reason in selection of organizational commitment is, we found a literature gap between Pygmalion effect and organization commitment as referred in our work.

Reviewer 2 Report

Comment on Economies-1686722.

Abstract

  1. Rewrite your abstract in a concise manner. For example,

“The study examines the Pygmalion effect and leader-member exchange on employee task performance via organizational commitment. Data were collected from AAA employees in BBB country. There are three research findings. First, CCC. Second, DDD. Third, EEE.

Theoretical part  

  1. The introduction needs to be improved. What is your research theme? If it is Pygmalion effect, focus on this theme. After you put forward this research theme, briefly review the work in this area. What are the antecedents and outcome variables explored by prior studies? Then, point out problems (limitations, weaknesses, or gaps) in the current literature. Given these problems, how will you address them? State their research purposes and objectives and theoretical perspective in support of your framing. Finally, articulate your research contributions.
  2. In your current writing, you seem to introduce each construct, without properly putting them together in the logical connection. Please consider point 2 to strength and streamline your introduction.
  3. You need to fucus on your work in the framework of Pygmalion effect on task performance. You mention too many constructs beyond the framework.
  4. For the literature review and hypotheses, you do not need to introduce each construct one by one but integrate them for the argumentation of hypotheses. I would suggest rewriting the part. First of all, briefly introduce the Pygmalion effect. Then, use the theoretical perspective to argue for your hypotheses.
  5.  For the hypotheses, as noted, focus on Pygmalion effect. I have some suggestions, just for your reference.

Hypothesis 1: Leader expectations are positively related to LMX (H1a, see the study by Bezuijen, van den Berg, van Dam, & Thierry, 2009) and job engagement (H1b).

Hypothesis 2: Leaders’ expectations are positively related to employee task performance.

Hypothesis 3: LMX (H3a) and job engagement (H3b) mediate the relationship between leaders’ expectations and task performance.

  1. You can also consider the moderating hypothesis. For example,

Hypothesis 1: LMX moderates the relationship leaders’ expectations and employee job engagement such that the relationship is stronger for those with higher level of LMX. 

Hypothesis 2: LMX moderates the indirect effect of leaders’ expectations on task performance via employee job engagement such that the indirect effect is stronger for those with higher level of LMX.

  1. Please reconsider whether organizational commitment is the most suitable mediator for the Pygmalion effect. As point 7 suggests, employee job engagement could be a better mediator than organizational commitment. You need to have a thorough review of prior studies on Pygmalion effect and build your model on the basis of these prior studies.

Empirical part

  1. You should avoid single-source bias (collecting data solely from one source). Employee task performance could be evaluated by supervisors.
  2. I would also suggest collecting data of all the variables from different time points. For example, employees report supervisor expectations at Time 1, their organizational commitment, and job engagement at Time 2, and supervisors evaluate subordinate performance at Time 3.
  3. Current Table 1, Table 2 and Table 3 are not necessary. You can use words to report the response rate.
  4. In your new Table 1, you can report means, standard deviations, and correlations among variables.
  5. In your new Table 2, you can report results of comparisons of measurement models (alternative models that combine any two variables).
  6. To report the mediation effect or indirect effect, you should offer the estimate as well as 95% CI. I do not see the 95%CI.
  7. It is a bit strange that you add T-values on the model path of Figure 2 on page 10.
  8. Carefully report the measures of variables, including source (authors and publication year), number of items, range scale, sample items, and alpha reliability.
  9. Comment on Discussion part will be given after the above concerns are properly addressed.

Author Response

Abstract

(1) We rewrote abstract in a more concise manner to the suggested pattern.

Theoretical part

(2) In theoretical background, we have highlighted the research gap and focused on Pygmalion effect variable; the rest of the variables are explained as supporting variables. (3) Then, the constructs were put in a systematic and logical sequence. (4,5) Additionally, we decided not to introduce our constructs one by one but combined them together with our hypotheses’ arguments as suggested. (6) The hypotheses were reorganized to a suggested pattern (for example, H1a, H1b etc.). (7) In context of your valuable suggestion to moderate hypotheses, unfortunately we cannot add at this stage because that will modify our results. However, we are considering this moderation in our coming study. (8) The reason in selection of organizational commitment is, we found a literature gap between Pygmalion effect and organization commitment as referred in our work.

Empirical part

(9) We already considered this point and we collected data from middle-level managers; this point was already justified in previous studies. (10) The survey data collection took almost 11 months. We assume that this time period fulfils the different time requirements. (11) We agree that some of our tables (old Tables 1-3) are not necessary and were integrated into the plain text. (12) At the same time, we added means values, standard deviations and (14) 95% confidence interval in our table with results (new Table 3) as suggested. (13) We explained the measurement model value in text. (15) We also removed T-values on the model path from Figure 2 as proposed. (16) We also reported the measures of variables carefully. (17) We would also appreciate any comments on the discussion part as well as comments to the changes that have been made so far.

Round 2

Reviewer 1 Report

The paper is now ready for publication in my opinion.

Reviewer 2 Report

Comment economies-1686722-v2

  1. As your model is similar to that of Wang, Law, Hackett, Wang and Chen (2005, AMJ), I would suggest closely following that study in revising your work.
  2. Revise your hypotheses as follows.

Hypothesis 1: Leaders’ expectation is positively related to employee task performance.

Hypothesis 2: Leader-member exchange is positively related to employee task performance.

Hypothesis 3: Organizational commitment relates positively to employee task performance.

Hypothesis 4a: Organizational commitment mediates the relationship between leaders’ expectation and employee task performance.

Hypothesis 4b: Organizational commitment mediates the relationship between leader-member exchange and employee task performance.

  1. Please write your Sample and Procedure and Measures according to Wang, Law, Hackett, Wang and Chen (2005, AMJ).
  2. Please use tables and the figure in Wang, Law, Hackett, Wang and Chen (2005, AMJ) as your tables and figure. You can keep your Table 1 and replace Table 2 of the AMJ paper.